# Concurrent coverage and determinants of vitamin A supplementation and deworming among children aged 12–59 months in 15 Sub-Saharan African countries

Thomas Kidanemariam Yewodiaw[1]*, Mequanint Dessie Bitewa[2], Abiyu Abadi Tareke[3]*

**1** Medical Officer at International Medical Corps, Amhara Region Emergency Operation Center, Gondar Field Office, Gondar, Ethiopia, **2** Department of Public Health, College of Health Sciences, Debre Markos University, Debre Markos, Ethiopia, **3** Nossal Institute for Global Health, Melbourne School of Population and Global Health, University of Melbourne, Level 2, Carlton, Victoria, Australia

* thomaskmariam28@gmail.com (TKY); abiyu2010@gmail.com (AAT)

## Abstract

### Background

Vitamin A supplementation (VAS) and deworming (DW) are proven, cost-effective interventions that protect children against preventable morbidity, mortality, and the burden of micronutrient deficiencies and parasitic infections. Nevertheless, many children fail to receive both interventions simultaneously, limiting the potential health gains. Assessing co-coverage and its determinants is crucial for guiding integrated child health strategies and closing persistent gaps across Sub-Saharan Africa.

### Methods

We analyzed DHS data from 15 Sub-Saharan African countries, including 107,725 children aged 12–59 months. The primary outcome was co-coverage of vitamin A supplementation and deworming within six months. Weighted descriptive statistics and mixed-effects logistic regression assessed determinants at individual, household, community, and country levels, accounting for survey design and clustering. Variables with $p < 0.20$ or deemed theoretically relevant were included in the multilevel model ($p < 0.05$, 95% CI).

### Results

The pooled co-coverage of vitamin A supplementation (VAS) and deworming (DW) among children aged 12–59 months was 44.0% (95% CI: 43.4–44.6%), despite individual coverage of 57.1% for each intervention. Approximately 13% of children received either Vitamin A or deworming as a single intervention. Nearly 30% of children received neither intervention. Co-coverage was lowest in Sierra Leone (10.3%)

**Data availability statement:** This study is based on publicly available Demographic and Health Survey (DHS) data. The datasets analyzed are available from the DHS Program website (https://dhsprogram.com/data/ ) upon reasonable request after registration and approval of data access.

**Funding:** The author(s) received no specific funding for this work.

**Competing interests:** No competing interest.

and Gabon (13.2%), moderate in Burkina Faso (28.6%), Côte d'Ivoire (31.0%), Mozambique (44.0%), and Tanzania (44.7%), and highest in Lesotho (58.4%) and Rwanda (84.7%). Vitamin A supplementation coverage was lowest in Gabon (15.7%), Sierra Leone (16.5%) and highest in Rwanda (89.1%) and Lesotho (73.6%), while deworming alone was lowest in Sierra Leone (30.3%) and Burkina Faso (36.7%) and highest in Rwanda (89.4%) and Lesotho (62.4%). Co-coverage of vitamin A supplementation and deworming was higher among children aged 24–47 months (AOR = 1.07), fully immunized children (AOR = 1.41), and those with older, educated mothers who attended antenatal care (AORs 1.14–1.59) or had media exposure (AOR = 1.13). Household wealth also increased the likelihood (AORs 1.27–1.64), while urban residence reduced it (AOR = 0.84). At the country level, compared with Burkina Faso, Rwanda (AOR = 20.05), Mauritania (AOR = 4.30), and Lesotho (AOR = 3.92) had the highest odds, whereas Gabon (AOR = 0.36) and Sierra Leone (AOR = 0.22) had the lowest inter-country disparities in integrated child health coverage. The intraclass correlation coefficient (ICC) indicated that approximately 21.6% of the variance in concurrent coverage was attributable to between-country differences.

## Conclusion

Co-coverage of vitamin A supplementation and deworming in 15 sub-Saharan Africa countries is low, with only 44% of children aged 12–59 months receiving both interventions, far below the WHO 80% target. Coverage varied widely, with Rwanda leading and Sierra Leone and Gabon lagging. Strengthened harmonized campaigns, routine service integration, and targeted outreach are essential to improve equitable child health outcomes.

---

## Introduction

Vitamin A deficiency (VAD) and soil-transmitted helminths (STH) remain co-endemic among children aged 12–59 months in Sub-Saharan Africa (SSA), causing morbidity, mortality, and the burden of micronutrient deficiencies and parasitic infections [1,2]. The World Health Organization (WHO) recommends giving vitamin A twice a year, together with yearly deworming (DW). Doing both at the same time rather than separately helps boost children's immunity, reduce parasite infections, and make programs more efficient [3–5]. VAD and infections caused by STH remain major global health problems, affecting approximately 190 million and 1.5 billion children worldwide, respectively [2,6]. In Sub-Saharan Africa (SSA), vitamin A deficiency remains a major public health concern, affecting approximately one-third to nearly half of preschool-aged children. Similarly, STH infections, particularly Ascaris lumbricoides, are highly prevalent [7,8], with an estimated 28–30% of children infected. These conditions contribute substantially to adverse health outcomes, including anemia, impaired immunity, and growth retardation [9]. Despite longstanding policies, coverage remains heterogeneous within SSA countries, with gaps driven by

socioeconomic inequities, geographic access, variable campaign performance, and service fragmentation [10–12]. Concurrently delivering vitamin A and deworming via Child Health Days or EPI touchpoints boosts coverage and cuts costs, but success hinges on coordination, supply reliability, and community engagement [4,7,10]. Looking at the proportion of children who receive both vitamin A supplementation and deworming on schedule gives a clearer picture of how well programs are working and highlights gaps in integrated delivery [11–13]. Co-coverage determinants operate across levels: child age, maternal (age, education, media exposure, employment), household wealth, and community factors (rurality, region, outreach intensity, facility access [11–14]. The DHS offers standardized, nationally representative data that let us track how many children receive both interventions and explore the factors behind it across countries. This helps identify gaps in equity and guide policies for more effective integrated delivery [13,15]. Accordingly, this study combines recent DHS data from sub-Saharan Africa to measure how many children aged 12–59 months receive both VAS and DW, and to identify individual, household, and community factors that can help improve integrated child health programs.

## Methods

### Study design and data sources

We used data from the most recent DHS conducted between 2019 and 2024 in 15 sub-Saharan Africa (SSA) countries. The DHS employs a stratified two-stage cluster sampling design that yields nationally representative data on women, children, and households. The analysis focused on 107,725 children aged 12–59 months living with their mothers at the time of the survey. Although VAS is recommended beginning at 6 months of age, DW is administered starting at 12 months; therefore, children younger than 12 months were excluded to ensure comparability of co-coverage for both interventions. For this analysis, we drew on the Kids Recode files, which provide information on children under five years of age and their mothers. We restricted the sample to children aged 12–59 months, as this age group is eligible for both VAS and DW interventions according to WHO guidelines. Delivery Platform Profiles: To contextualize VAS and DW co-coverage, we collected information on each country's delivery platform during the DHS survey year, drawing from UNICEF program reports, Ministry of Health guidelines, and published literature. Delivery mechanisms were categorized as routine facility-based delivery, campaign-based delivery (e.g., biannual Child Health Days), or mixed platforms combining routine services with community outreach or periodic campaigns [16–21]. These profiles were compiled into a Supplementary Table (Table 2) and used to interpret cross-country differences in co-coverage. For example, Mozambique and Kenya primarily used routine delivery with active community health worker outreach, while Tanzania relied on biannual Child Health Days with limited routine VAS. Incorporating delivery platform data allowed us to better understand structural drivers of low, moderate, and high co-coverage patterns.

**Study population.** The analytic sample consisted of 107,725, 12–59-month-old children who had valid information on VAS and DW in the six months preceding the survey. Children with missing or "don't know" responses were excluded from the analysis. Pooling across countries provided adequate statistical power to assess both individual- and community-level determinants while capturing cross-country variations.

**Outcome variable.** The primary outcome was co-coverage, defined as receipt of both VAS and DW within the six months preceding the survey, based on maternal recall and verified with child health cards when available. Children who received both interventions were classified as having concurrent coverage. VAS and DW captured by DHS variables, each recoded as binary (1 = received, 0 = not received). A composite variable was created: 0 = neither VAS nor DW received, 1 = received either VAS or DW only, and 2 = received both (full co-coverage). For regression, the outcome was further binary (1 = co-coverage, 0 = not co-coverage) [22,23].

**Independent variables.** We examined factors influencing co-coverage at three interconnected levels: individual, household, and community. At the individual level, we considered the child's age, sex, birth order, immunization status, and maternal characteristics, including age, education, employment status, antenatal care (ANC) visits, and exposure to

media [3,24–34]. At the household level, we assessed the family's wealth index, household size, and place of residence (urban or rural) [35,36]. At the community level, we aggregated data on maternal education, household wealth, and media exposure within each cluster, categorizing them as "low" or "high" based on national medians. This approach aligns with methodologies used in previous research to understand community-level influences on child health [33,37].

## Statistical analysis

All analyses accounted for the complex survey design by applying DHS sampling weights, primary sampling units, and strata using the svy commands in Stata. In bringing together data from the 15 countries, we wanted to look at the results from two different angles. The first was by using population-proportionate weights, which keep each country's share of the data in line with its actual population size. This way, larger countries naturally contribute more to the overall estimate, making the pooled result closer to the true picture of the whole region. The second approach was to use equal-country weights, where every country counts the same, no matter its population. This method gives smaller countries an equal voice in the analysis, making comparisons across countries fairer and not dominated by the bigger ones. Similar strategies have been used in past multi-country DHS studies to balance representativeness with comparability. Descriptive statistics were used to summarize background characteristics and the coverage of VAS, DW, and their co-coverage. Bivariate associations were tested using chi-square statistics.

To examine determinants of co-coverage, we fitted multilevel mixed-effects logistic regression models (melogit in Stata), recognizing the hierarchical nature of the data (children nested within clusters, and clusters nested within countries). The fixed effects estimated associations between predictors and co-coverage, while random effects captured between-cluster and between-country variability. Adjusted odds ratios (AOR) with 95% confidence intervals (CI) were reported. Model fit was assessed using the intraclass coefficient (ICC), Akaike information criterion (AIC), and Bayesian Information Criterion (BIC).

## Ethical considerations

This study is a secondary analysis of publicly available data from the most recent DHS conducted between 2019 and 2024 across multiple Sub-Saharan Africa (SSA) countries. The original DHS surveys obtained ethical approval from the ICF Institutional Review Board and the respective national ethics committees of each participating country. Written informed consent was obtained from all respondents during the survey interviews. The datasets contain no personal identifiers, thereby ensuring participant confidentiality. Access to the data was granted by the DHS Program under their standard data use agreement (https://dhsprogram.com).

## Results

### Characteristics of the study population

The analysis included 107,725 children aged 12–59 months, with a fairly even distribution across age groups: 27,687 (25.7%) were 12–23 months, 26,120 (24.5%) were 24–35 months, 27,409 (25.4%) were 36–47 months, and 26,509 (24.3%) were 48–59 months. Slightly more children were female (54,473; 50.4%) than male (53,252; 49.6%). About two-thirds of children were second-born or higher: 39,996 (38.5%) were second- or third-born, and 42,326 (35.5%) were fourth-born or higher.

Most mothers were aged 25–34 years (50,746; 47.4%), with 27,199 (24.9%) aged 15–24 years and 29,780 (27.8%) aged 35–49 years. Regarding maternal education, 41,277 (29.5%) had no formal education, 34,366 (33.9%) had primary education, 27,462 (30.1%) had secondary education or higher, and 4,620 (6.5%) had tertiary education. Children were from households across all wealth levels: 30,350 (22.9%) in the poorest quintile and 14,612 (18.0%) in the richest quintile. Most children lived in rural areas (70,904; 59.6%) compared to urban areas (36,821; 40.4%). The sample spanned

15 Sub-Saharan countries, with Kenya (14,679; 18.2%), Mauritania (8,569; 12.8%), Madagascar (9,071; 7.0%), Tanzania (7,998; 6.8%), and Ghana (7,052; 6.6%) contributing the largest shares. Other countries each accounted for smaller proportions ranging from 3.0% to 6.0% of the sample (Table 1).

**Table 1. Sociodemographic characteristics of children aged 12–59 months and their households across 15 sub-Saharan Africa countries (weighted).**

| Characteristic | Category | N | % (weighted) |
|---|---|---|---|
| **Child age (months)** | 12–23 | 27,687 | 25.7 |
| | 24–35 | 26,120 | 24.5 |
| | 36–47 | 27,409 | 25.4 |
| | 48–59 | 26,509 | 24.3 |
| **Child sex** | Male | 53,252 | 49.6 |
| | Female | 54,473 | 50.4 |
| **Birth order** | 1 | 25,403 | 26.0 |
| | 2–3 | 39,996 | 38.5 |
| | ≥4 | 42,326 | 35.5 |
| **Maternal age (years)** | 15–24 | 27,199 | 24.9 |
| | 25–34 | 50,746 | 47.4 |
| | 35–49 | 29,780 | 27.8 |
| **Maternal education** | No education | 41,277 | 29.5 |
| | Primary | 34,366 | 33.9 |
| | Secondary+ | 27,462 | 30.1 |
| | Tertiary | 4,620 | 6.5 |
| **Household wealth quintile** | Poorest | 30,350 | 22.9 |
| | Poorer | 23,087 | 19.2 |
| | Middle | 21,304 | 20.0 |
| | Richer | 18,372 | 19.9 |
| | Richest | 14,612 | 18.0 |
| **Place of residence** | Urban | 36,821 | 40.4 |
| | Rural | 70,904 | 59.6 |
| **Country/Region** | Burkina Faso | 9,188 | 5.5 |
| | Côte d'Ivoire | 7,646 | 5.8 |
| | Gabon | 4,473 | 4.2 |
| | Gambia | 5,887 | 3.0 |
| | Ghana | 7,052 | 6.6 |
| | Kenya | 14,679 | 18.2 |
| | Lesotho | 1,768 | 4.2 |
| | Liberia | 3,925 | 3.5 |
| | Madagascar | 9,071 | 7.0 |
| | Mauritania | 8,569 | 12.8 |
| | Mozambique | 6,725 | 6.6 |
| | Rwanda | 6,198 | 5.4 |
| | Senegal | 6,877 | 6.2 |
| | Sierra Leone | 7,669 | 4.3 |
| | Tanzania | 7,998 | 6.8 |

*N* = unweighted frequency; *% (weighted)* = weighted percentage using DHS sampling weights.

**Source:** Demographic and Health Surveys (DHS), 2018–2023.

### Comparative coverage of Vitamin A supplementation, deworming, and co-coverage of 15 sub-Saharan Africa countries

Overall, pooled concurrent coverage of VAS and DW among 107,725 children aged 12–59 months was 44.0% (47,599 children; 95% CI: 43.4–44.6). Individually, 57.1% of children (61,503; 95% CI: 56.4–57.7) received vitamin A, and the same proportion received deworming (61,503; 95% CI: 56.5–57.7). Notably, 13.1% of children (14,106) received only one of the interventions, while 55.9% (60,126 children; 95% CI: 55.4–56.6) received either one or neither, highlighting substantial gaps in integrated service delivery and missed opportunities for maximizing child health benefits.

Co-coverage of both interventions, which means the proportion of children who received vitamin A and deworming concurrently, was highest in Rwanda (84.7%), followed by Lesotho (58.4%), Senegal (56.6%), and Mauritania (54.5%). In contrast, the lowest co-coverage was reported in Sierra Leone (10.3%) and Gabon (13.2%), reflecting differences delivery of the two interventions. Countries such as Mozambique (44%), Tanzania (44.7%), and Kenya (49.5%) showed moderate co-coverage, while Ghana, despite its high vitamin A coverage, showed lower co-coverage (37.7%) due to modest deworming uptake.

The coverage of VAS and DW among children aged 12–59 months varied widely across the 15 Sub-Saharan Africa countries. VAS ranged from as low as 15.7% in Gabon and 16.5% in Sierra Leone to as high as 89.1% in Rwanda. Other countries with relatively high VAS coverage included Ghana (74.3%), Lesotho (73.6%), and Senegal (68.4%). In contrast, Burkina Faso (41.5%), Madagascar (39.2%), and Côte d'Ivoire (46.1%) revealed moderate uptake.

DW coverage showed a similar heterogeneity, with the highest levels observed in Rwanda (89.4%), Gabon (67.9%), and Senegal (69.2%). Conversely, the lowest coverage was documented in Sierra Leone (30.3%), Burkina Faso (36.7%), and Côte d'Ivoire (46.9%). Some countries, such as Ghana (44.6%) and Mozambique (47.5%), recorded moderate uptake despite having relatively strong VAS. These findings show substantial cross-country variation, with some countries achieving strong and balanced delivery of both interventions (Rwanda, Senegal, and Lesotho), while others show programmatic gaps where one intervention far outpaces the other (Gabon with deworming, Ghana with vitamin A) (Fig 1).

### Co-coverage of vitamin A supplementation and deworming delivery platforms by Country

Co-coverage of VAS and DW varies widely across the 15 countries, reflecting differences in delivery platforms and program coordination. Rwanda achieved the highest co-coverage (84.7%), followed by Lesotho (58.4%) and Tanzania (44.7%), illustrating the effectiveness of well-coordinated campaigns combined with routine delivery. Countries with mixed platforms, such as Kenya, Senegal, Mozambique, Ghana, and The Gambia, attained moderate co-coverage (37–56%), where routine and campaign approaches improved access, but inconsistencies in timing or targeting limited simultaneous uptake. Conversely, Sierra Leone, Gabon, Burkina Faso, Côte d'Ivoire, and Madagascar exhibited low co-coverage (10–31%) due to fragmented, poorly documented, or unsynchronized delivery systems. These findings underscore that high individual coverage of either intervention does not automatically translate into high co-coverage; achieving optimal simultaneous uptake requires careful integration, timing coordination, and alignment of delivery platforms (Table 2) [18,20,21,38–44].

### Determinants of vitamin A supplementation and deworming, and co-coverage in sub-Saharan Africa countries

Compared with children aged 12–23 months, those aged 24–35 months (AOR = 1.07; 95% CI: 1.03–1.12) and 36–47 months (AOR = 1.07; 95% CI: 1.02–1.13) had significantly higher odds, while children aged 48–59 months were less likely to be covered (AOR = 0.92; 95% CI: 0.88–0.97). Child's sex and birth order showed no consistent associations.

Children of mothers aged 25–34 (AOR = 1.14; 95% CI: 1.09–1.19) and 35–49 (AOR = 1.25; 95% CI: 1.18–1.33) were more likely to receive the intervention compared to younger mothers (15–24 years). Education showed a clear gradient: primary (AOR = 1.18; 95% CI: 1.13–1.23), secondary (AOR = 1.34; 95% CI: 1.27–1.40), and higher (AOR = 1.48; 95% CI: 1.36–1.62) all increased odds compared with no schooling.

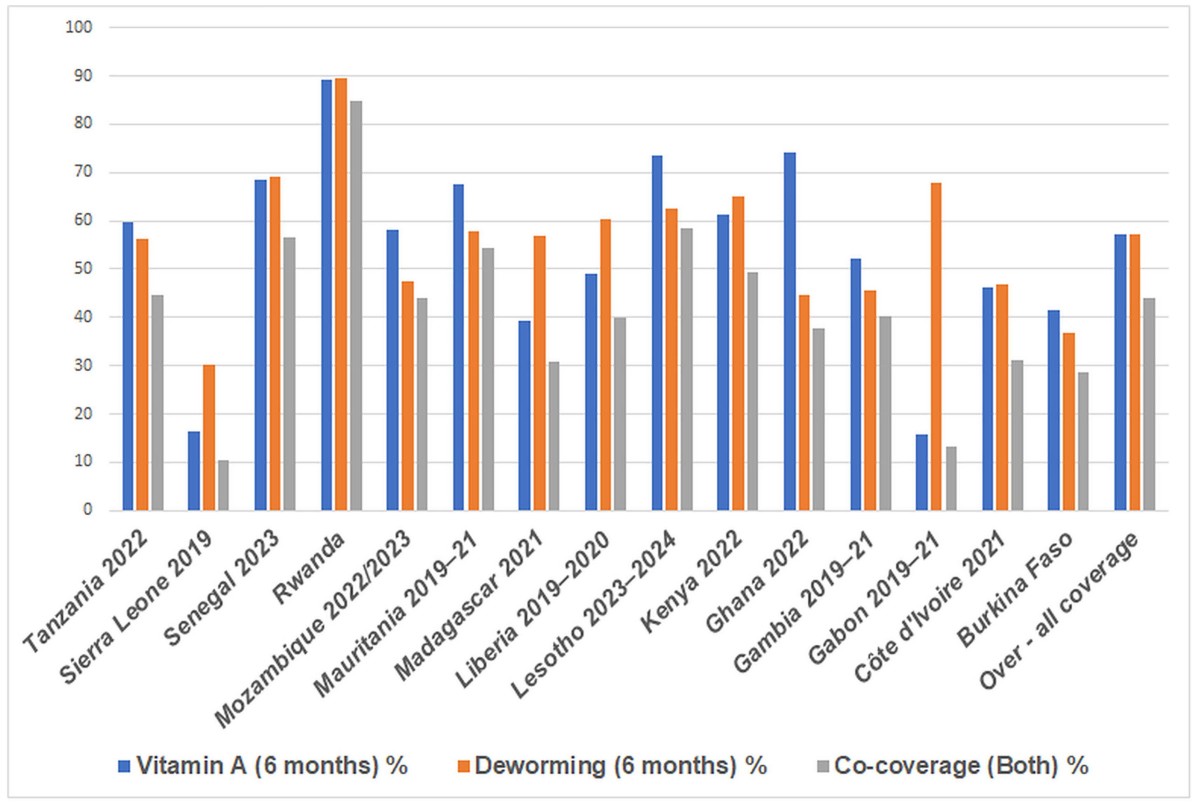

**Fig 1. Comparative coverage of vitamin A supplementation, deworming, and co-coverage among children aged 12–59 months in 15 Sub-Saharan Africa Countries, 2019–2024 DHS data.**

Compared with the poorest households, children in poorer (AOR = 1.14; 95% CI: 1.08–1.20), middle (AOR = 1.27; 95% CI: 1.20–1.34), richer (AOR = 1.42; 95% CI: 1.33–1.52), and richest (AOR = 1.64; 95% CI: 1.51–1.78) households had progressively higher odds. Media exposure was also significant (AOR = 1.13; 95% CI: 1.07–1.19).

Fully immunized children were more likely to receive co-coverage (AOR = 1.41; 95% CI: 1.35–1.48). Antenatal care visits also mattered to mothers with 1–3 visits (AOR = 1.50; 95% CI: 1.34–1.68) and 4 + visits (AOR = 1.59; 95% CI: 1.43–1.77) had greater odds than those with no ANC.

Children from medium (AOR = 1.26; 95% CI: 1.17–1.36) and high (AOR = 1.33; 95% CI: 1.21–1.47) cluster wealth areas were more likely to be covered. Higher cluster education was also beneficial, with medium (AOR = 1.22; 95% CI: 1.14–1.30) and high (AOR = 1.29; 95% CI: 1.19–1.40) levels increasing odds compared with low-education clusters. Urban residence was associated with reduced odds (AOR = 0.84; 95% CI: 0.78–0.91).

Finally, relative to Burkina Faso, countries such as Côte d'Ivoire (AOR = 1.37), Gambia (AOR = 2.02), Kenya (AOR = 2.53), Lesotho (AOR = 3.92), Mauritania (AOR = 4.30), Mozambique (AOR = 2.20), Rwanda (AOR = 20.05), Senegal (AOR = 4.70), Tanzania (AOR = 2.13), and Liberia (AOR = 2.16) had significantly higher odds of co-coverage. In contrast, Gabon (AOR = 0.36) and Sierra Leone (AOR = 0.22) had markedly lower odds. Stepwise inclusion of individual, community, and country-level factors reduced clustering from ICC 37.3% (MOR 3.79) in the null model to 21.6% (MOR 2.48) in the full model, explaining 53.7% of the between-cluster variance (PCV). The final model showing the best fit (lowest AIC/BIC) (Table 3).

Table 2. Delivery platforms for co-coverage vitamin A supplementation and deworming and their co-coverage implications by country.

| Country | VAS delivery platform | Deworming delivery platform | Key source(s)/ notes |
|---|---|---|---|
| Burkina Faso | Campaign/ Child Health Event (twice-annual) | Campaign co-delivery with VAS during rounds | UNICEF program & country reports show two rounds/ VAS Days. (UNICEF Open) [1] |
| Côte d'Ivoire | Mixed/transitioning toward routine (documented routine pilots & routine-strengthening) | Campaigns for deworming documented; platform for under-5s mixed/ unclear | Evaluation of transition to routine in Côte d'Ivoire (HKI/ UNICEF Delivery Effectiveness brief). (Health Campaign Effectiveness Coalition) [2] |
| Gabon | Unknown/ no clear public program doc located — confirm with MOH/UNICEF | Unknown | No clear, dated public document found in indexed searches — recommend country confirmation. |
| The Gambia | Mixed/ campaign rounds + routine support (national nutrition mapping shows rounds and routine support) | Campaign co-delivery documented; routine support mentioned | National Nutrition mapping & UNICEF Gambia materials. (unnutrition.org.org) [3] |
| Ghana | Mixed — routine + periodic mass distributions in some regions | School-based for school-age + campaigns/outreach for pre-school in some regions | Program evaluations & UNICEF/ NI notes indicate mixed approaches. (Scaling Up Nutrition) [4,5] |
| Kenya | Mixed — Malezi Bora/MNCH Weeks + routine & CHV outreach | Mixed: Malezi Bora/mass events + school-based for older children | Documentation on Malezi Bora and CHV outreach; program evaluations. (UNICEF) [3] |
| Lesotho | Routine immunization platform + outreach (documented in UNICEF Lesotho programmed docs) | Routine + outreach/ integrated with immunization contacts | UNICEF Lesotho programmed materials note the use of immunization/outreach as a platform. (Health Campaign Effectiveness Coalition) [6] |
| Liberia | Unknown/public documentation unclear (NTD partners deliver deworming MDAs historically) | Deworming: MDA/NTD campaigns by partners historically; platform for 6–59m unclear | NTD partner activity documented; VAS platform at DHS time not clearly documented — confirm with MOH/UNICEF [7]. |
| Madagascar | Unknown/ no clear public program doc located in indexed searches | Unknown | Recommend checking MOH/ UNICEF country office records for the DHS period. |
| Mauritania | Unknown/ no clear publicly indexed doc located | Unknown | Confirm with MOH/UNICEF [8]. |
| Mozambique | Routine + CHW outreach/ integrated with primary care (not heavily campaign-dependent) | Routine + outreach/ integrated (campaigns used in some contexts) | Program literature and reviews show routine/integrated emphasis. (PMC) |
| Rwanda | Mixed — documented campaign rounds in recent years + routine | Campaign rounds + routine; school-based for older children | GAVA/UNICEF program notes show recent campaign rounds (e.g., 2021) and routine elements. (MedRxiv) [9] |
| Senegal | Mixed/ shifting toward routine & community-based delivery; campaigns used in some districts | Campaigns & school-based deworming; routine elements were scaled up | UNICEF and program reviews indicate moves toward routine delivery with catch-up campaigns. (UNICEF) [10] |
| Sierra Leone | Unknown | Unknown | Historically, NTD/deworming MDAs present; the VAS platform during the DHS period was not clearly documented or confirmed with MOH/UNICEF. |
| Tanzania | Campaign/ Biannual Child Health Days (CHDs) | Campaign/ CHDs commonly co-deliver VAS + deworming | Tanzania MOH program descriptions & UNICEF/ NI reporting on CHDs. (UNICEF Open) [9] |

## Discussion

### Pooled concurrent coverage of vitamin A supplementation and deworming among children in 15 sub-Saharan Africa countries

In this study, only 44.0% of children aged 12–59 months received VAS and DW concurrently, highlighting a substantial gap in co-coverage despite relatively high individual intervention rates. This gap indicates missed opportunities for integrated service delivery, which is essential to maximize child health benefits and program efficiency [32,45]. Ensuring co-coverage is particularly important because concurrent delivery can have synergistic effects on nutritional status, immunity, and growth outcomes [27,32]. Individual coverage of VAS and DW was higher, at 57.1% for each intervention, suggesting that delivery platforms

**Table 3. Multilevel mixed-effects logistic regression of determinants of child co-coverage in Sub-Saharan Africa Countries, 2019–2024.**

| Variable | Category | Model 1 (AOR: CI) | Model 2 (AOR: CI) | Model 3 (AOR: CI) | Model 4 (AOR: CI) |
|---|---|---|---|---|---|
| **Sex** | Female | | 0.98 (0.95–1.01) | | 0.98 (0.95–1.01) |
| **Child age (ref: 12–23)** | 24–35 | | 1.08 (1.03–1.12) | | 1.07 (1.03–1.12) |
| | 36–47 | | 1.1 (1.05–1.16) | | 1.07 (1.02–1.13) |
| | 48–59 | | 0.95 (0.9–1.0) | | 0.92 (0.88–0.97) |
| **Mother's age (ref: 15–24)** | 25–34 | | 1.19 (1.13–1.24) | | 1.14 (1.09–1.19) |
| | 35–49 | | 1.34 (1.26–1.42) | | 1.25 (1.18–1.33) |
| **Mother's education (ref: None)** | Primary | | 1.31 (1.25–1.36) | | 1.18 (1.13–1.23) |
| | Secondary+ | | 1.38 (1.32–1.45) | | 1.34 (1.27–1.40) |
| | Higher | | 1.54 (1.41–1.68) | | 1.48 (1.36–1.62) |
| **Wealth index (ref: Poorest)** | Poorer | | 1.19 (1.13–1.25) | | 1.14 (1.08–1.20) |
| | Middle | | 1.39 (1.32–1.46) | | 1.27 (1.20–1.34) |
| | Richer | | 1.60 (1.5–1.7) | | 1.42 (1.33–1.52) |
| | Richest | | 1.88 (1.75–2.01) | | 1.64 (1.51–1.78) |
| **Media exposure** | Yes | | 1.08 (1.02–1.13) | | 1.13 (1.07–1.19) |
| **Birth order (ref: 1)** | 2–3 | | 0.98 (0.94–1.02) | | 1.00 (0.96–1.05) |
| | 4+ | | 0.94 (0.89–0.99) | | 0.99 (0.93–1.04) |
| **Fully immunized** | Yes | | 1.45 (1.39–1.52) | | 1.41 (1.35–1.48) |
| **ANC visits (ref: None)** | 1–3 visits | | 1.55 (1.39–1.74) | | 1.50 (1.34–1.68) |
| | 4 + visits | | 1.51 (1.35–1.68 | | 1.59 (1.43–1.77) |
| **Cluster wealth (ref: Low)** | Medium | | | 1.48 (1.38–1.58) | 1.26 (1.17–1.36) |
| | High | | | 1.90 (1.74–2.09) | 1.33 (1.21–1.47) |
| **Cluster education (ref: Low)** | Medium | | | 1.31 (1.23–1.41) | 1.22 (1.14–1.30) |
| | High | | | 1.51 (1.39–1.63) | 1.29 (1.19–1.40) |
| **Cluster media (ref: Low)** | Medium | | | 1.03 (0.97–1.10) | 0.98 (0.92–1.04) |
| **Residence (ref: Rural)** | Urban | | | 0.89 (0.82–0.95) | 0.84 (0.78–0.91) |
| **Country (ref: Burkina Faso** | Côted'Ivoire | | | 1.28 (1.11–1.48) | 1.37 (1.19–1.58) |
| | Gabon | | | 0.36 (0.30–0.43) | 0.36 (0.30–0.44) |
| | Gambia | | | 2.08 (1.76–2.46) | 2.02 (1.71–2.39) |
| | Ghana | | | 1.57 (1.36–1.82) | 1.47 (1.27–1.70) |
| | Kenya | | | 2.73 (2.42–3.08) | 2.53 (2.24–2.86) |
| | Lesotho | | | 4.20 (3.50–5.05) | 3.92 (3.26–4.71) |
| | Liberia | | | 2.11 (1.78–2.49) | 2.16 (1.83–2.55) |
| | Madagascar | | | 1.09 (0.95–1.26) | 1.10 (0.95–1.26) |
| | Mauritania | | | 4.15 (3.65–4.71) | 4.30 (3.78–4.88) |
| | Mozambique | | | 2.22 (1.92–2.56) | 2.20 (1.91–2.54) |
| | Rwanda | | | 23.1 (19.7–27) | 20.05 (17.1–23.4) |
| | Senegal | | | 4.6 (4–5.3) | 4.70 (4.08–5.41) |
| | Sierra Leone | | | 0.22 (0.18–0.26) | 0.22 (0.18–0.26) |
| | Tanzania | | | 2.2 (1.94–2.56) | 2.13 (1.85–2.44) |

successfully reach a majority of children with at least one service [45]. However, the difference between individual coverage and co-coverage underscores the limitations of current delivery strategies in achieving synchronized uptake [27,32,45]. Evidence from several countries highlights the advantages of integrating VAS with DW. Studies from Bangladesh and Indonesia show that children who received both β-carotene (a precursor of vitamin A) and deworming experienced greater improvements

in serum retinol and growth outcomes than those who received either intervention alone [45–47]. Similarly, research from Zaire reported that VAS improved growth among vitamin A-deficient children, whereas deworming alone offered minimal benefits [46,47]. These findings support the potential of combined interventions to enhance child health [27,32,46,47]. Yet, the evidence is not entirely consistent. Systematic reviews and meta-analyses, including Cochrane reviews, indicate that deworming alone often yields variable effects on growth, cognition, and nutritional outcomes [48,49]. In Uttar Pradesh, India, twice-yearly deworming among lightly infected children did not significantly improve weight gain or reduce mortality [27]. Some studies suggest that gains in growth and survival are largely driven by VAS rather than deworming, particularly in areas with low infection intensity [46–48]. These mixed results may explain the persistently low co-coverage, highlighting that the impact of integrated delivery depends on context-specific factors such as nutritional status, infection burden, and intervention timing [48,50]. Targeting combined delivery to high-impact settings is key to maximizing child health benefits.

Programmatically, co-delivery of vitamin A and deworming is safe and feasible, with no adverse interactions reported between vitamin A and albendazole [51]. Yet, operational challenges often limit coverage. In Siaya Country, Kenya, VAS reached over 80% of children, while deworming reached only 50% [52]. Similarly, in South Africa, many children missed both interventions despite high vaccination coverage [53]. Supply chain issues, differing campaign schedules, and workforce constraints are likely key contributors to these gaps [52,53].

Overall, VAS and DW are widely recognized as safe, essential, and compatible interventions [27,45,52]. They can offer added benefits, particularly in areas where vitamin A deficiency and helminth infections overlap [27,46–48]. Yet, evidence shows that deworming alone often has a limited impact, and the success of combined delivery can vary depending on context [45,48]. Efforts to improve co-coverage should focus on integrated planning, coordinated delivery, and targeted outreach to high-risk populations, taking local epidemiology and health system capacity into account [32,50,51].

## Comparative analysis of co-coverage of vitamin A supplementation and deworming across sub-Saharan Africa countries

High co-coverage countries, including Rwanda (84.7%), Lesotho (58.4%), Senegal (56.6%), and Mauritania (54.5%), demonstrate successful integration of VAS and DW programs. Rwanda stands out with VAS and DW coverage above 89%, reflecting strong health systems, well-trained community health workers, and coordinated campaigns. Lesotho and Senegal have also achieved co-coverage above 50% through national coordination and effective community engagement [50,53]. These findings align with global evidence that well-planned, integrated campaigns enhance simultaneous uptake, maximize health benefits, and improve cost-effectiveness [50,54]. These countries could achieve Sub-Saharan Africa (SSA) substantial co-coverage through campaign-based approaches, such as biannual Child Health Days, allowing simultaneous provision of VAS and DW with strong partner coordination [55,56].

Moderate co-coverage countries, such as Mozambique (44.0%), Tanzania (44.7%), Kenya (49.5%), and Ghana (37.7%), show partial success. In Mozambique, VAS and deworming reach children through routine, outreach, and occasional campaigns, but poor coordination means many children miss receiving both. Kenya demonstrates better synchronization, while Tanzania shows high VAS but lower deworming coverage. These patterns reflect partial integration, often caused by staggered campaign schedules, workforce limitations, and regional inequities [34,51,52]. This pattern aligns with evidence that partial integration usually occurs when one intervention reaches children effectively while the other lags behind [48,51]. Programmatic strategies such as synchronized delivery, stronger community engagement, and better resource allocation have been shown to improve co-coverage in such contexts [34]. Moderate co-coverage highlights room for improvement, with operational gaps in synchronized campaigns, monitoring, and community outreach [48,51]. Those countries rely on routine facility-based delivery supplemented by community health worker outreach, which provides continuous access but is constrained by supply-chain challenges and inconsistent campaign alignment [17,55].

Low co-coverage countries, including Sierra Leone (10.3%) and Gabon (13.2%), highlight persistent delivery challenges. In Gabon, VAS coverage was only 15.7% despite 67.9% deworming coverage, while Sierra Leone had moderate

deworming (30.3%) but very low VAS uptake (16.5%. These low rates are consistent with studies identifying logistical constraints, unsynchronized campaigns, weak community mobilization, and supply chain interruptions as major barriers [27,45,48]. Systematic planning, aligned campaigns, and strengthened health systems are essential to improve co-coverage [45,48]. Overall, co-coverage varies widely, with high-performing countries benefiting from strong governance and coordinated campaigns, while moderate and low performers face operational and systemic barriers. Targeted strategies focusing on synchronized delivery, high-risk populations, and health system strengthening are critical to maximize child health outcomes across Sub-Saharan Africa [34,45–54]. Low co-coverage countries face systemic barriers, while high performers offer models of effective integration. Strengthening co-coverage will require adopting best practices such as coordinated campaigns, robust supply chains, and active community engagement [53]. Categorizing countries by co-coverage allows targeted strategies to optimize child health outcomes across Sub-Saharan Africa. Low-coverage countries often have fragmented delivery systems, with VAS and DW delivered separately through vertical programs or irregular campaigns, resulting in poor simultaneous uptake [52,56]. Differences in co-coverage were closely aligned with the underlying delivery platforms. Countries relying predominantly on campaign-based delivery, such as Tanzania, often achieved higher peak coverage but showed variability between rounds. In contrast, settings where routine delivery with community outreach is the primary platform (Mozambique, Kenya) demonstrated more stable but moderate coverage, often constrained by health worker shortages and geographic access challenges. These platform differences highlight that co-coverage performance cannot be interpreted solely from DHS data without considering operational delivery models.

## Determinants of integrated vitamin A supplementation and deworming coverage in children

Child age showed a mixed pattern: children between 24–47 months were more likely to receive both interventions, while those aged 48–59 months were less likely. This suggests that coverage is higher in early childhood but declines as children grow older, likely due to fewer contact opportunities with health services. Similar trends have been noted in programmatic evaluations of Child Health Days and routine service platforms [3,30]. Maternal age showed a clear positive gradient, with older mothers more likely to ensure their children received both interventions. Similar findings have been reported elsewhere, linking maternal maturity with better care-seeking behaviors and greater ability to navigate health services [28,29]. Maternal education was also one of the strongest predictors, showing a dose–response effect from primary through higher education. This aligns with extensive evidence that maternal literacy improves uptake of immunization, supplementation, and other preventive child health practices [22,26]. Household wealth showed a clear gradient, with children from the richest households nearly twice as likely to receive both interventions compared to those from the poorest. This reflects the persistent pro-rich inequalities seen in the use of preventive child health services across Sub-Saharan Africa [31,57]. Media exposure also played an important role, as mothers with access to radio or television were more likely to ensure their children received both interventions. This supports earlier evidence that information exposure through mass media improves maternal health service use and uptake of child health interventions [24,33]. Children who were fully immunized had substantially higher odds of co-coverage, underlining the importance of integrated service delivery platforms such as routine immunization and campaign contacts [57,58]. Antenatal care (ANC) utilization proved to be a strong predictor of co-coverage. Mothers who attended 1–3 visits, and particularly those with four or more, were more likely to have their children receive both interventions. This aligns with the continuum of care framework, which emphasizes how maternal engagement with health services directly supports the uptake of preventive child health interventions [25,28,29]. Community wealth and education were positively associated with co-coverage, even after adjusting for individual factors, though the effect was reduced in the full model. In contrast, community-level media exposure was not significant once household media access was accounted for, highlighting the stronger influence of direct information exposure on child health service uptake [31,35]. Unexpectedly, children living in urban areas were less likely to receive both interventions. This contrasts with the usual urban advantage. In many countries, this pattern aligns with existing evidence showing that rural outreach campaigns and Child Health Days often target hard-to-reach communities, resulting

in relatively higher coverage in rural areas. More consistent community health worker (CHW) outreach in rural settings, congestion within urban health systems, and variability in the reach of campaign-based platforms may also contribute to these differences [30]. This study's main strengths are the use of large, nationally representative DHS data from 15 Sub-Saharan Africa countries and the focus on concurrent coverage of VAS and DW, which provides multi-country insights into overlapping child health interventions. The cross-sectional design limits causal inference, and supplementation and DW data relied on maternal recall, partially mitigated by verification with health cards. DHS surveys do not capture supply-chain conditions; while VAS capsules are generally well supplied through global programs, DW tablets are often scarce, and unmeasured stock-outs may have influenced DW and co-coverage estimates. Additionally, programmatic variables such as campaign timing and input availability were not captured, limiting the assessment of operational factors affecting coverage.

## Conclusions

Less than half of children aged 12–59 months in Sub-Saharan Africa receive both vitamin A supplementation and deworming, well below the WHO-recommended 80% target, despite higher individual coverage. High-performing countries, such as Rwanda and Lesotho, have exceeded targets through coordinated campaigns, while moderate and low-coverage countries, including Mozambique, Sierra Leone, and Gabon, face operational and systemic barriers. Uptake is strongly influenced by maternal age, education, household wealth, antenatal care attendance, and child immunization. Low-coverage countries require strengthened routine delivery and reliable deworming supply chains; moderate-coverage settings need improved supervision, microplanning, and community–facility linkages; and high-coverage countries demonstrate the effectiveness of synchronized platforms. Enhancing coordination, supply chains, and integration with routine services can improve efficiency, equity, and child health outcomes across the region.

## Author contributions

**Conceptualization:** Thomas Kidanemariam Yewodiaw, Abiyu Abadi Tareke.

**Data curation:** Thomas Kidanemariam Yewodiaw, Mequanint Dessie Bitewa, Abiyu Abadi Tareke.

**Formal analysis:** Thomas Kidanemariam Yewodiaw.

**Investigation:** Thomas Kidanemariam Yewodiaw, Abiyu Abadi Tareke.

**Methodology:** Thomas Kidanemariam Yewodiaw, Mequanint Dessie Bitewa, Abiyu Abadi Tareke.

**Project administration:** Thomas Kidanemariam Yewodiaw.

**Resources:** Thomas Kidanemariam Yewodiaw.

**Software:** Thomas Kidanemariam Yewodiaw, Mequanint Dessie Bitewa, Abiyu Abadi Tareke.

**Supervision:** Mequanint Dessie Bitewa, Abiyu Abadi Tareke.

**Validation:** Mequanint Dessie Bitewa, Abiyu Abadi Tareke.

**Visualization:** Mequanint Dessie Bitewa, Abiyu Abadi Tareke.

**Writing – original draft:** Thomas Kidanemariam Yewodiaw.

**Writing – review & editing:** Thomas Kidanemariam Yewodiaw, Mequanint Dessie Bitewa, Abiyu Abadi Tareke.

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
