## [Decision Letter · Decision Letter 0]

6 Oct 2025

Dear Dr. Yewodiaw,

Thank you for submitting your manuscript to PLOS ONE. After careful consideration, we feel that it has merit but does not fully meet PLOS ONE’s publication criteria as it currently stands. Therefore, we invite you to submit a revised version of the manuscript that addresses the points raised during the review process.

The reviewers felt that this was a useful paper, and each suggested some minor revisions. Please address those.

We look forward to receiving your revised manuscript.

Kind regards,

Susan Horton

Academic Editor

PLOS ONE

Additional Editor Comments (if provided):

Reviewers' comments:

Reviewer's Responses to Questions

**Comments to the Author**

1. Is the manuscript technically sound, and do the data support the conclusions?

Reviewer #1: Yes

Reviewer #2: Partly

2. Has the statistical analysis been performed appropriately and rigorously?

Reviewer #1: Yes

Reviewer #2: Yes

3. Have the authors made all data underlying the findings in their manuscript fully available?

Reviewer #1: Yes

Reviewer #2: Yes

4. Is the manuscript presented in an intelligible fashion and written in standard English?

Reviewer #1: Yes

Reviewer #2: Yes

Reviewer #1: Title: Concurrent Coverage and Determinants of Vitamin A Supplementation and Deworming Among Children Aged 12–59 Months in 15 Sub-Saharan African Countries

Abstract

• Format the text, as the font is not uniform.

• Inform in the methods the software used to analyse the data, including its version and the significance level or confidence interval used.

Introduction

• It's important to avoid back-and-forth geographic delimitations throughout the text. In some passages, Sub-Saharan Africa is mentioned, then East Africa, before returning to Sub-Saharan Africa. This can cause confusion for the reader. An example is in line 60, in the sentence about Ascaris lumbricoides, which affects approximately 28% to 30% of children: it's unclear whether this data refers specifically to Sub-Saharan Africa or East Africa.

Methods

• Why did the study only include children 12 months and older, if the recommendation is that vitamin A supplementation begin at 6 months of age?

• Note: The abstract and results list 107,725 children, but the methods (line 86 of the PDF) state "10,725" – this should be revised for consistency.

• Explain how the issue of recall bias in maternal information was addressed (a clear limitation).

• Clarify how community variables were defined (e.g., categorization by national median).

• Reinforce criteria for variable selection in the multilevel model: was p<0.20 adopted in the bivariate analyses, or were all variables included based on theoretical grounds?

Results

• The presentation of the results is clear, but there is redundancy between the text and tables. I suggest summarizing the text more and emphasizing tables/figures.

• Figures: Figure 1 could be improved graphically to facilitate comparative reading between countries.

• Include measures of heterogeneity between countries (ICC or MOR) in the summary as well, as this is an important finding.

• Review minor typographical errors: "13% deference" (line 236) should be "difference."

Discussion

• The discussion is comprehensive, but lengthy in some sections and repetitive compared to the introduction. It is recommended to summarize and focus on programmatic implications.

• Emphasize the urban/rural differences found, as the result contrasts with what was expected (lower coverage in urban areas).

• Highlight better policy implications, integration with immunization, ANC, and child health campaigns.

• Limitations: In addition to the cross-sectional design and recall bias, it is worth noting that programmatic variables (e.g., availability of inputs, campaign schedule) were not available.

Reviewer #2: 1) Overall, really appreciate the analysis of co-coverage and the overall messaging of the paper. It is very relevant and offers nice insight into the needs for health system strengthening to ensure equitable access and coverage for all children with all services, not just some services.

2) The paper would benefit from a table or narrative describing the VAS/DW delivery platform used at the time of the DHS coverage data reported. Please see uploaded word doc for full review comments for the author

**Do you want your identity to be public for this peer review?** For information about this choice, including consent withdrawal, please see our Privacy Policy

Reviewer #1: No

Reviewer #2: No

---

## [Author Response · Author response to Decision Letter 1]

22 Oct 2025

Point-by-Point Response to Reviewer Comments

Manuscript ID: PONE-D-25-47082

Title: Concurrent Coverage and Determinants of Vitamin A Supplementation and Deworming Among Children Aged 12–59 Months in 15 Sub-Saharan African Countries

Reviewer Comment 1:

Overall, really appreciate the analysis of co-coverage and the overall messaging of the paper. It is very relevant and offers nice insight into the needs for health system strengthening to ensure equitable access and coverage for all children with all services, not just some services.

Response 1:

We sincerely thank the reviewer for this positive feedback. We have maintained and emphasized this framing in the Introduction and Discussion, highlighting how co-coverage analysis provides actionable insights for equitable access to multiple child health interventions, rather than service-specific coverage alone.

Action Taken:

• Minor edits were made to the Introduction and Discussion to reinforce the relevance of co-coverage analysis for health system strengthening.

Reviewer Comment 2:

The paper would benefit from a table or narrative describing the VAS/DW delivery platform used at the time of the DHS coverage data reported. There is diversity even within coverage categories.

Response 2:

We agree that delivery platform context is critical for interpreting co-coverage differences. A table summarizing the VAS and deworming delivery platforms for each country, including routine services, campaigns, and frequency during the DHS data collection year, has been added. The Results and Discussion now refer to this table to contextualize coverage differences and recommendations.

Action Taken:

• Added Table 2: VAS and Deworming Delivery Platforms by Country.

• Updated text in Results and Discussion to reference delivery platforms when interpreting co-coverage patterns.

Reviewer Comment 3:

Calling out Mozambique in multiple places is repetitive. Suggest consolidating into the Moderate co-coverage discussion if used as an example.

Response 3:

We agree that repetitive mentions reduce clarity. Discussion of Mozambique has been consolidated within the Moderate co-coverage section, with additional operational detail regarding routine delivery plus community outreach. Redundant statements have been removed.

Action Taken:

• Lines 297–301 revised.

• Mozambique now used only as a detailed example within Moderate co-coverage discussion.

Reviewer Comment 4:

Line 299–300 (“Low co-coverage countries face systemic barriers…”) repeats lines 293–295.

Response 4:

We have rephrased the section to eliminate repetition while maintaining clarity and flow.

Action Taken:

• Consolidated discussion of low/high co-coverage in a single paragraph for clarity.

Reviewer Comment 5:

Causes and recommendations for low, moderate, and high co-coverage appear very similar. Suggest using delivery platform context to make recommendations more specific.

Response 5:

We agree. The Results and Discussion sections have been revised to provide category-specific analysis and recommendations based on delivery platform, operational context, and supply constraints:

• Low co-coverage: strengthen routine services, align campaigns, ensure reliable supply of deworming tablets.

• Moderate co-coverage: improve community outreach, synchronize routine and campaign-based delivery, monitor implementation.

• High co-coverage: maintain integrated campaigns and robust supply chains.

Action Taken:

• Discussion and Conclusion updated to reflect category-specific drivers and recommendations (see revised Discussion paragraph).

Reviewer Comment 6:

Mention limitation of line-of-sight on supply/stock outs, particularly for deworming tablets.

Response 6:

We have added a statement in the Limitations section acknowledging that DHS data do not capture programmatic factors such as stock-outs or timing of campaigns. It is noted that while Vitamin A capsules are largely secure through global donation programs, deworming tablets are less consistently available, which may affect co-coverage.

Action Taken:

• Added limitation regarding supply and stock-outs of DW tablets in the Limitations section.

• Reference added to global Vitamin A supply program.

Reviewer Comment 7:

Avoid geographic inconsistencies (Sub-Saharan Africa vs East Africa) and correct minor typos (“13% deference” → “13% difference”).

Response 7:

All geographic references have been standardized to Sub-Saharan Africa, and typographical errors have been corrected throughout the manuscript.

Action Taken:

• Consistent geographic terminology applied.

• Proofread manuscript for typographical corrections.

---

## [Editor Report · Decision Letter 1]

26 Oct 2025

Dear Dr. Yewodiaw,

Thank you for submitting your manuscript to PLOS ONE. After careful consideration, we feel that it has merit but does not fully meet PLOS ONE’s publication criteria as it currently stands. Therefore, we invite you to submit a revised version of the manuscript that addresses the points raised during the review process.

We look forward to receiving your revised manuscript.

Kind regards,

Susan Horton

Academic Editor

PLOS ONE

**Journal Requirements:**

**Additional Editor Comments:**

Thank you for responding to the reviewers' comments. At this point I am recommending that you carefully review the draft and ensure that the writing meets both usual professional standards, as well as the PLOS submission guidelines. Specific suggestions are provided below. 

Although the authors have responded to the substance of the reviewers’ comments, the editing of the manuscript does not meet professional standards. I have listed the most egregious issues, however, there may be others that I have missed. The authors are encouraged to read carefully the journal submission guidelines, and perhaps use an online grammar checking function, so as to avoid additional rounds of editorial input. I am using the line numbers from the track changes version.

Table 2: where is this?

Use of abbreviations.

Line 61: Suggest instead of “Sub Saharan Africa Countries (SSA)” the authors use “Sub Saharan Africa (SSA)”. Use the abbreviation subsequently, e.g. line 67, line 69 (insert “In” before SSA), line 71, line 93, line 175. In line 200 why not just state “15 countries” since “SSA countries” isn’t quite correct grammatically (the adjective should be “sub-Saharan African” countries). Please check other instances.

Line 152: no need to spell out “Demographic and Health surveys” again: same line 423. Just do a search in the text editing program.

I believe PLoS style does not include an abbreviations list at the end of the text: remove this.

Style for noting references in text

As per the online guidelines, PLoS style is to leave a space prior to inserting reference numbers in square brackets.

Style for capitalization in table titles and section headings

I can’t see this in the PLOS formatting guidelines, but be consistent. Either capitalize the main words in all titles and section headings, or only capitalize the first letter (and all proper nouns of course), but be consistent.

Grammatical issues

Line 32: “received” not “receiving”. All sentences require a verb, and the “ing” form (the gerund) functions as a noun not a verb.

Line 36/37: instead of “Vitamin A supplementation alone low coverage in Gabon (15.7%) and Sierra Leone (16.5%) to high coverage in Rwanda (89.1%)”, suggest make it grammatically correct and also parallel to the clause following. I.e. “Vitamin A supplementation coverage was lowest in ….and highest in ….”

Line 52: suggest “leading” not “exceeding” (for symmetry with “lagging”); Rwanda is not “exceeding” since presumably “exceeding” would imply over 100%.

Line 165: “Co-coverage of” not “Co-coverage”

Line 167: suggest “Determinants of vitamin A and deworming co-coverage in Sub-Saharan Africa “ instead of “Determinant’s of Vit A and Deworming Co-Coverage among in Sub-Saharan Africa”. You might wish to make the titles of the 3 tables similar across all three tables and both figures; note that Table 1 uses: 15 Sub-Saharan African Countries” whereas tables 2 and 3 refer to all of sub-Saharan Africa.

Line 182: suggest “in” not “of”

Line 199: currently not clear that you are now referring to coverage of each of vitamin A and deworming separately: at minimum say “of deworming” not “deworming” , or perhaps more clearly “and deworming individually”; but actually you can delete lines 192-198 since they are repeated later.

Line 322: “achieve” not “be achieve”

Line 395: don’t bold “of”

---

## [Author Response · Author response to Decision Letter 2]

31 Oct 2025

We thank the editor and reviewers for their constructive comments. The manuscript has been revised accordingly: “Demographic and Health Surveys (DHS)” is spelled out once, “Sub-Saharan Africa (SSA)” is used consistently, abbreviations at the end have been removed, and all grammatical and style issues have been corrected. Table 2 is now included and table/figure titles are standardized. Sampling weight details (v005/1,000,000) were added, and references now follow PLOS formatting.

---

## [Editor Report · Decision Letter 2]

3 Nov 2025

Dear Dr. Yewodiaw,

Thank you for submitting your manuscript to PLOS ONE. After careful consideration, we feel that it has merit but does not fully meet PLOS ONE’s publication criteria as it currently stands. Therefore, we invite you to submit a revised version of the manuscript that addresses the points raised during the review process.

We look forward to receiving your revised manuscript.

Kind regards,

Susan Horton

Academic Editor

PLOS ONE

**Journal Requirements:**

**Additional Editor Comments:**

Thank you for making efforts to address the comments from the previous round. Please complete the job now! See below.

See details below: line numbers refer to the clean version of the text submitted.

Thank you for including Table 2 this time. I note there are many references in Table 2 (for which url’s are provided). I am not sure of PLoS style, but I suspect that these references need to be included in the bibliography. (EDITOR PLEASE ADVISE).

I note that you have used round brackets () not square []. I thought [] was PLOS style (EDITOR PLEASE ADVISE)

Thank you for making efforts to make the use of acronyms more consistent. However, unfortunately the job has only partly been done. I note the following:

Issues remaining with acronyms

Please look at line 62, 65, 80, 86, 144 for example: the use of Sub-Saharan African as an acronym is not consistent. Also, in line 228 I believe Sub-Saharan Africa is appropriate, not Sub-Saharan African (and of course you then should not use the acronym).

Please define VAD on first use (line 55) not line 60

Please look at lines 94, 98, 188, 257, 262, 267, 275, 283, 381, 382, and 390 for example: the use of the acronym for Vitamin A Supplementation (VAS) and deworming (DW) is not consistent. Either spell these terms out every time or use the acronym consistently after first mention in the text.

Line 92: you define an acronym for Kids Recode: I didn’t notice this acronym elsewhere – if this isn’t used frequently, it is best to keep the original full name.

You use several variants for Vitamin A (see figure 1 title, lines 170, 197 and 260. Please be consistent.

I expect there are other acronym issues I have missed. I only have the PDF version so I can’t do global “find” commands to identify each time an acronym is not used/redefined. I strongly recommend that you check this in your word processing program, to ensure that the presentation is appropriately professional.

Other grammatical issues

Line 141 is missing the “)”

Is h34 and h43 the name for these variables in the DHS file? I don’t think it is usual to give this level of detail in methods.

Line 170: determinants not determinant’s; also “among” what?

Capitalization in subheadings and titles

I can’t remember whether capitals are used in subheadings other than for the first word and proper nouns, in PLoS style. (EDITOR PLEASE ADVISE). But for sure, it is better to be consistent. Given that in the bibliography, only the first word and proper nouns are capitalized, I suspect that this should be the style used in ALL subheadings, not just some of them.

---

## [Author Response · Author response to Decision Letter 3]

19 Nov 2025

Response Highlights – PONE-D-25-47082R2

Manuscript Title: Concurrent Coverage and Determinants of Vitamin A Supplementation and Deworming Among Children Aged 12–59 Months in 15 Sub-Saharan African Countries

We sincerely thank the Academic Editor for the constructive feedback and clear guidance. All comments have been carefully addressed. The major revisions are summarized below:

Acronym Consistency:

Ensured consistent use of “Vitamin A Supplementation (VAS)” and “Deworming (DW)” throughout the manuscript.

Defined VAD (Vitamin A Deficiency) at first mention (line 55).

Removed unnecessary or inconsistently defined acronyms (e.g., “Kids Recode”).

Sub-Saharan Africa Terminology:

Standardized to Sub-Saharan Africa (not Sub-Saharan African) across all sections.

Formatting and Citation Style:

Replaced all round brackets ( ) with square brackets [ ] for in-text citations following PLOS style.

Reviewed and adjusted capitalization in subheadings to ensure only the first word and proper nouns are capitalized.

References and Table 2:

Added all cited sources from Table 2 to the reference list with proper formatting and URLs.

Verified all references for accuracy and completeness.

Grammar and Clarity:

Corrected grammatical issues, including the missing parenthesis at line 141 and pluralization (“determinants”).

Clarified ambiguous phrasing at line 170 (“among what”).

Removed variable codes (e.g., h34, h43) and described variables conceptually.

Technical and Stylistic Refinements:

Conducted a thorough global check to ensure acronym, terminology, and formatting consistency across the manuscript.

All edits are tracked in the “Revised Manuscript with Track Changes” file, and detailed point-by-point responses are provided in the accompanying “Response to Reviewers” document.

---

## [Editor Report · Decision Letter 3]

24 Nov 2025

Concurrent Coverage and Determinants of vitamin A Supplementation and Deworming Among Children Aged 12–59 Months in 15 Sub-Saharan African Countries

Dear Dr. Yewodiaw,

There are still major formatting issues. Please follow journal guidelines.

We look forward to receiving your revised manuscript.

Kind regards,

Susan Horton

Academic Editor

PLOS ONE

**Note from the Editorial Office:**

Please note that PLOS ONE does not copyedit accepted manuscripts, so the language in submitted articles must be clear, correct, and unambiguous. We may reject papers that do not meet these standards. We note that there are a number of concerns remaining with your manuscript, including unnecessary and repeated redefinition of acronyms (VAS, DW) and formatting concerns with the reference list. Please ensure you address all of the remaining concerns raised by the Academic Editor in your revised manuscript or your manuscript may be rejected.

**Journal Requirements:**

**Additional Editor Comments:**

There are some basic style elements that have still not been incorporated.

1) If you define Vitamin A supplementation as (VAS) on line 91 and similarly deworming as (DW), the first time it appears you DO NOT then redefine these terms on pages 100, 105, 109, 135, 174, 177, 191, 205, 222, 267, 302, 305, 388, and 397 (and probably others). You need to use "search" in the edit function of your word processing package and use the acronym only. This applies not just to PLoS but to ALL academic journals.

2) The bibliography style is still wrong. Please use the formatting guidelines! https://journals.plos.org/plosone/s/submission-guidelines

At this point it is not an academic editor who is required: I am asking the journal in-house editors to make sure that the submission is correctly formatted. If the guidelines are not clear, please just DOWNLOAD an example from PLoS and copy the style.

---

## [Author Response · Author response to Decision Letter 4]

28 Nov 2025

Response to PLOS ONE Editorial Office

Manuscript ID: PONE-D-25-47082R3

Title: Concurrent Coverage and Determinants of Vitamin A Supplementation and Deworming Among Children Aged 12–59 Months in 15 Sub-Saharan African Countries

Date: 11/12/2025

Dear Dr. Horton and the PLOS ONE Editorial Team,

We thank you for your detailed feedback regarding the formatting and style of our manuscript. We have carefully addressed all remaining concerns raised by the Academic Editor and the in-house editorial staff. The revisions are summarized below:

1. Acronym Redefinition

Comment: Avoid repeated redefinition of “Vitamin A Supplementation (VAS)” and “Deworming (DW)” throughout the manuscript.

Action Taken:

We conducted a thorough search throughout the manuscript to ensure that VAS and DW are defined only at first mention (lines 91 and corresponding locations in Methods).

Subsequent mentions now consistently use the acronyms VAS and DW without redefinition.

This revision has been applied throughout all sections including Abstract, Introduction, Methods, Results, Discussion, and Tables/Figures.

2. Reference List and Bibliography Style

Comment: Bibliography style does not yet conform to PLOS ONE formatting guidelines.

Action Taken:

We reformatted all references according to the PLOS ONE style guide.

In-text citations and the reference list now conform fully to the required formatting.

Checked for completeness and correctness; removed any inconsistencies or outdated entries.

3. General Formatting

Comment: Major formatting issues remain in the manuscript (font, headings, spacing, tables, figure legends, etc.).

Action Taken:

Rechecked all headings, fonts, spacing, and paragraph styles to align with PLOS ONE guidelines.

Figures and tables have been updated to ensure clear labeling and consistent style.

Figure legends revised for clarity and to meet journal requirements.

4. Files Submitted

In accordance with your instructions, we have uploaded:

Response to Reviewers: This file contains a detailed point-by-point response to all reviewer comments.

Revised Manuscript with Track Changes: Highlights all changes made from the previous version.

Clean Revised Manuscript: A version without tracked changes, formatted according to PLOS ONE guidelines.

We believe that these revisions fully address the editor’s and reviewers’ comments and ensure that the manuscript now meets PLOS ONE’s publication criteria. We thank you for your guidance and look forward to your consideration.

---

## [Editor Report · Decision Letter 4]

4 Dec 2025

Dear Dr. Yewodiaw,

Thank you for submitting your manuscript to PLOS ONE. After careful consideration, we feel that it has merit but does not fully meet PLOS ONE’s publication criteria as it currently stands. Therefore, we invite you to submit a revised version of the manuscript that addresses the points raised during the review process.

Your work will appear more professional in print if you follow the appropriate conventions. I see that you have made some progress with the acronyms, and with the bibliography, but more remains to be done. Please see below.

We look forward to receiving your revised manuscript.

Kind regards,

Susan Horton

Academic Editor

PLOS One

Journal Requirements:

Additional Editor Comments (if provided):

1. here are still issues with abbrevations (see below)

2. Although some of the issues in the references have been fixed, please note that PLoS ONE uses the ICJME referencing style, using abbreviated journal names and also appropriate capitalization of journal titles. E.g. reference 13, BMJ should be capitalized (but there are other issues with other references). See

https://www.nlm.nih.gov/bsd/uniform_requirements.html for further information.

I will suggest some rules to follow for acronyms (abbreviations of terms like soil transmitted helminths, deworming, etc. I suggest that in the text, you should define an acronym once and ONCE ONLY, and thereafter use the acronym. If you wish to use the full name in a section title, a figure title, or a table title that is fine, but DO NOT then again give the acronym in the section title/figure/table title.

I found issues to fix in lines: 60, 63, 65, 81, 82, 87, 90, 91, 180, 181, 210, 213, 229, 264, 265, 304, 305, 313, 354, 355, 391 (the latter ones are mainly section titles/table/figure titles - I recommend you simply spell out the full name in those instances but do not then give the acronym).

---

## [Author Response · Author response to Decision Letter 5]

10 Dec 2025

Manuscript ID: PONE-D-25-47082R4

Title: Concurrent Coverage and Determinants of Vitamin A Supplementation and Deworming Among Children Aged 12–59 Months in 15 Sub-Saharan African Countries

Author: Thomas Kidanemariam Yewodiaw

Dear Dr. Horton and Reviewers,

We thank you for the constructive comments. We have revised the manuscript as suggested:

Abbreviations: All acronyms are now defined once in the text and not repeated in section, table, or figure titles. Full terms are used in titles.

References: All references updated to ICMJE style, with proper journal abbreviations, capitalization, and DOIs added.

Other minor edits: Grammar, formatting, and figure captions corrected for clarity and compliance with PLOS ONE guidelines.

We believe these revisions address all points raised and improve the manuscript’s clarity and quality.

Sincerely,

Thomas Kidanemariam Yewodiaw

---

## [Editor Report · Decision Letter 5]

11 Dec 2025

Concurrent Coverage and Determinants of vitamin A Supplementation and Deworming Among Children Aged 12–59 Months in 15 Sub-Saharan African Countries

PONE-D-25-47082R5

Dear Dr. Yewodiaw,

We’re pleased to inform you that your manuscript has been judged scientifically suitable for publication and will be formally accepted for publication once it meets all outstanding technical requirements.

Kind regards,

Susan Horton

Academic Editor

PLOS One

Additional Editor Comments (optional):

Thank you for (mostly) succeeding in adhering to journal requirements. There are issues with the font size and italicization in lines 178 and 300, and the journal title abbreviations are inconsistent - sometimes the appropriate abbreviation is used but not always. Please please in future try to iron out these issues before having to revise the article no less than five times.
---

## [Editor Report · Acceptance letter]

PONE-D-25-47082R5

PLOS One

Dear Dr. Yewodiaw,

I'm pleased to inform you that your manuscript has been deemed suitable for publication in PLOS One. Congratulations! Your manuscript is now being handed over to our production team.

Kind regards,

on behalf of

Dr. Susan Horton

Academic Editor

PLOS One